# Alteration of Community Metabolism by Prebiotics and Medicinal Herbs

**DOI:** 10.3390/microorganisms11040868

**Published:** 2023-03-28

**Authors:** Christine Tara Peterson, Josué Pérez-Santiago, Stanislav N. Iablokov, Dmitry A. Rodionov, Scott N. Peterson

**Affiliations:** 1Center of Excellence for Research and Training in Integrative Health, Department of Family Medicine, School of Medicine, University of California San Diego, La Jolla, CA 92093, USA; 2Division of Cancer Biology, University of Puerto Rico Comprehensive Cancer Center, San Juan, PR 00927, USA; 3School of Dental Medicine, Office of Research, University of Puerto Rico Medical Sciences Campus, San Juan, PR 00921, USA; 4Phenobiome Inc., Palm Springs, CA 92262, USA; 5Bioinformatics and Structural Biology Program, Sanford Burnham Prebys Medical Discovery Institute, La Jolla, CA 92037, USA; 6Tumor Microenvironment and Cancer Immunology Program, Sanford Burnham Prebys Medical Discovery Institute, La Jolla, CA 92037, USA

**Keywords:** glycosyl hydrolases, glycan, gut microbiota, short-chain fatty acid, community metabolism, medicinal herb, prebiotic, prebiotics, genome-wide metabolic reconstruction

## Abstract

Several studies have examined the impact of prebiotics on gut microbiota and associated changes in host physiology. Here, we used the in vitro cultivation of human fecal samples stimulated with a series of chemically related prebiotics and medicinal herbs commonly used in Ayurvedic medicine, followed by 16S rRNA sequencing. We applied a genome-wide metabolic reconstruction of enumerated communities to compare and contrast the structural and functional impact of prebiotics and medicinal herbs. In doings so, we examined the relationships between discrete variations in sugar composition and sugar linkages associated with each prebiotic to drive changes in microbiota composition. The restructuring of microbial communities with glycan substrates alters community metabolism and its potential impact on host physiology. We analyzed sugar fermentation pathways and products predicted to be formed and prebiotic-induced changes in vitamin and amino acid biosynthesis and degradation. These results highlight the utility of combining a genome-wide metabolic reconstruction methodology with 16S rRNA sequence-based community profiles to provide insights pertaining to community metabolism. This process also provides a rational means for prioritizing in vivo analysis of prebiotics and medicinal herbs in vivo to test hypotheses related to therapeutic potential in specific diseases of interest.

## 1. Introduction

It has been estimated that ~90% of the nutrients and micronutrients ingested by humans are absorbed in the small intestine [1]. As a result, the vast and densely populated microbial communities in the large intestine and fecal microbiota have adapted a specialized functional capacity to scavenge more limited resources to support their energy and metabolic demands [2]. The selective and spatially segregated absorption of dietary nutrients and generalized loss of gylcosyl hydrolases (GHs) function in the human genome [3] select microbes in order to encode numerous and diverse GH functions to ensure efficient utilization of available energy sources that ultimately drive cooperative cross-feeding activities that maintain a stable microbial community structure and homeostasis of the human gut ecosystem. These features represent an important explanation for the growing focus among investigators in evaluating the impact of prebiotics and medicinal herbs on gut microbial communities and their ability to confer positive health benefits to the host.

The availability of dietary and host-derived oligosaccharides (the latter derived largely from the mucin layer) within the gut drives the selection for critical functions that contribute to the community architecture of the resident microbiota [3]. GH families encoded by the genomes of gut microbes range from small to vast, in several instances, encompassing a large fraction of each genome’s total coding potential. Specialists in glycan metabolism include members of the genera *Bacteroides, Parabacteroides, Prevotella,* and lactic acid bacteria that have undergone substantial evolutionarily tuning to acquire the machinery required for the metabolism of diverse dietary oligosaccharides. It may then seem surprising that many gut microbes do not encode substantial glycan catabolism potential. This apparent paradox may be explained by what is increasingly appreciated as a feature of complex bacterial communities that have evolved to establish cooperative non-overlapping functions and molecular cross-feeding. In this way, gut communities have arisen that are substantially more than the sum of their individual species members. The extra-cellular localization of many or most GHs allows specialist microbes to breakdown complex glycans and, importantly, to share the simple sugar products with GH “deficient” microbes endowed with sugar transport and utilization pathways.

The full functional diversity and catalytic specificities of carbohydrate-active enzymes encoded by gut microbiomes remains incompletely known. Functionally, glycan metabolism capacity in gut microbes is species- and strain-specific and many contain metabolic diversity for several carbon sources, as with most classes of oligosaccharides. For example, *Lactobacillus* spp. display metabolic diversity for glycan metabolism in line with the phylogenetic diversity within a group that has evolved towards a reduced genome size [4]. Oligosaccharide catabolic enzymes in Lactobacilli are almost exclusively intracellular and thus oligosaccharide transport is a limiting step in metabolism [5]. Examples of species with such metabolic diversity include *Lactobacillus acidophilus*, *L. plantarum*, and *L. casei*; however, a far narrower glycan fermentation potential is retained by other lactic acid bacteria such as *L. sanfranciscensis*, which retains preference for only two disaccharide substrates [6]. By contrast, other gut bacteria contain extensive functions for the extracellular hydrolysis of glycans and subsequent transport of oligosaccharides, disaccharides, and monosaccharides without metabolic restriction on intracellular functions [7]. These varied strategies in glycan metabolism reflect niche adaptation and fitness at multiple gastrointestinal sites. Thus, both broad and narrow catabolic and fermentative capacities exist within the species of the gastrointestinal ecosystem.

Some species residing in the gut are specialized to metabolize host glycans abundantly present and constantly renewed in the mucin layer. The communities inhabiting the specialized niche are likely to have evolved to retain catabolic pathways specific for the *O*-linked oligosaccharides presented by mucin [8]. The mucosal-associated communities display overlap with the composition of luminal communities but constitute distinct structures and cross-feeding behavior. Like colonic lumen-resident communities, mucin-foraging communities feature dominant taxa encoding expansive GH functions, and these taxa include *Streptococcus*, *Helicobacter*, *Akkermansia*, *Bacteroides*, *Bifidobacterium*, *Clostridium*, *Prevotella*, *Ruminococcus*, *Streptomyces*, which all encode mucin-degrading capacities [9].

The dominant gut genera *Bacteroides* and *Prevotella* encode a large set of overlapping carbohydrate active enzymes (CAZymes) as well as unique sets of GH families, suggesting common and distinct carbohydrate utilization capacities [10]. *Prevotella* spp. have broad functional utilization potential for foraging a variety of complex plant-derived glycans and are often in higher abundance in the context of human populations consuming plant-based diets, while *Bacteroides* species are associated with diets high in animal protein that are also glycosylated as glycoproteins [11]. Dietary modification with plant prebiotic fibers and medicinal herbs drives shifts in the glycan utilization potential of the gut microbiota [12,13,14]. This dietary response is individualized, featuring a higher metabolic competence of species in certain individuals. Importantly, the selective restructuring of microbial communities by glycans necessarily restructures community metabolism and the products of sugar and amino acid fermentation (short-chain fatty acids) as well as other biochemical processes. The ability to restructure community metabolism has significant therapeutic implications as a growing number of microbially derived metabolites are being documented as having a direct impact on host physiology and health. Among the best studied examples is the SCFA, a butyrate that improves gut motility and stimulates tolerance and anti-inflammatory pathways through the expansion of Tregs [15]. Microbially catalyzed tryptophan degradation generates a number of metabolites with therapeutic potential as these metabolites influence gut motility (5-HT), gut barrier integrity as a spectrum of indole metabolites [16], and they may positively affect mood and emotional features [17].

In this study, we analyzed a series of prebiotic fibers that are chemically and compositionally similar and distinct to dissect the impact that variations in sugar linkages and compositional complexity have on altering microbial community structure and function. These findings were compared to a diverse group of medicinal herbs reported by our group previously [14,18,19,20] in an attempt to better understand how features of glycans differentially impact community metabolism. In addition to the 16S rRNA sequence profiling of resulting communities, we applied genome-wide metabolic reconstruction to predict the effect of prebiotics and medicinal herbs in order to change the representation of GHs, SCFA, vitamins, and sugar utilization potential. The results of these analyses highlight a procedure to rationally select prebiotics and medicinal herbs for studies in animal and human models of disease to achieve maximal therapeutic impact.

## 2. Materials and Methods

***Study participants and sample collection.*** Healthy, English-speaking women and men aged 30–60 years that had previously adhered to a vegetarian or vegan diet for >1 year were recruited to donate a single stool sample. This study was carried out in accordance with the recommendations of the Sanford Burnham Prebys Medical Discovery Institute Institutional Review Board (IRB-2014-020) guidelines, with written informed consent from all subjects. All subjects gave written informed consent in accordance with the Declaration of Helsinki. The protocol was approved by the Sanford Burnham Prebys Medical Discovery Institute Institutional Review Board. Participants ate their normal diets and donated a morning fecal sample in stool hats (Fisher Scientific, La Jolla, CA, USA). The fecal samples were transferred to conical tubes and stored at −80 °C until subsequent processing.

***Anaerobic fecal cultures.*** Equal volumes of stool collected from 12 healthy vegetarian participants were pooled and used to inoculate (approximately 10^6^ cells) a chemically defined medium (CDM) or CDM supplemented with prebiotics or medicinal herbs (1% *w*/*v*). Anaerobic cultures (9% H_2_, 10% CO_2_, 81% N_2_) were grown statically for 2 days at 37 °C as technical replicates (*n* = 4–6) and grown to approximate saturation. Cultures were harvested after 48 h via centrifugation.

***Chemically defined media.*** The CDM contained 50 mM N-2-hydroxyethylpiperazine-N’-2-ethanesulfonic acid (HEPES), 2.2 mM KH_2_PO_4_, 10 mM Na_2_HPO_4_, 60 mM NaHCO_3_, 4 mM of each amino acid except leucine (15 mM), 10 mL ATCC, and a Trace Mineral Supplement. The CDM also contained nucleoside bases (100 mg/L), inosine, xanthine, adenine, guanine, cytosine, thymidine, and uracil (400 mg/L). Additionally, the CDM contained choline (100 mg/L), ascorbic acid (500 mg/L), lipoic acid (2 mg/L), hemin (1.2 mg/L), and myo-inositol (400 mg/L). Resazurin (1 mg/L) was added to visually monitor dissolved oxygen. The pHs of the media were adjusted to 7.4. The 2X CDM and prebiotics/medicinal herbs (powder) in sterile water (2%) were separately reduced in an anaerobic chamber (Coy Labs) for 2 days.

***Prebiotics and medicinal herbs.*** All prebiotics were sourced from Sigma-Aldrich (corn starch cat# S4126, potato starch cat# S2004, α cellulose cat# C8002, amylose potato cat# A0512, amylopectin corn cat# 10120, corn dextrin cat# D2006, apple pectin cat# 93854, citrus peel pectin cat# P9135, β-D glucan barley cat#G6513, inulin chicory cat#I2255, FOS cat#F8052, and gastric mucin porcine cat#M2378). All medicinal herbs were sourced from Banyan Botanicals (Albuqueque, NM, USA), except for Jatamansi that was obtained from AyurOrganics (Victoria, Australia).

***Microbial DNA isolation.*** Genomic DNA was isolated from SCFA-supplemented cultures using the procedures of the QiaAmp DNA stool kit (Qiagen) with a modification that included bead beating for 5 min with the Thermo FastPrep instrument (MP Bio, La Jolla, CA, USA) to ensure uniform lysis of bacterial cells.

**16S *rRNA sequence analysis***. Multiplexed 16S rRNA libraries were prepared using standard 16S rRNA metagenomic sequencing library protocols from Illumina, which uses the V3–V4 region of 16S rRNA sequencing for target amplification and subsequent analysis. We used Qiime 2 for all other taxonomic analyses at the species level and higher and for subsequent genome reconstruction. Briefly, raw sequence reads were filtered, denoised, paired-read merged, and chimeras removed using the default parameters in dada2 [21] to generate an abundance table with amplicon sequence variants (ASVs) representing individual 16S rRNA sequences. To assign taxonomic descriptions to the obtained ASVs we used the multi-taxonomy approach (MTA). Each ASV sequence was aligned with 16S rRNA sequences from the Ribosomal Database Project (RDP, version 11.5). The obtained alignments were sorted via the percent identity, with maximum values denoted as M. We further collected and processed taxonomic assignments for identified 16S rRNA sequences with identities higher than the M−(1−M)/4 threshold. Resulting multi-taxonomy assignments consisted of one or more taxonomic names separated by “/”. To account for variable 16S rRNA gene copy numbers in reference genomes, we further renormalized each sample`s ASV abundance by averaging 16S rRNA copy numbers at each taxonomic level provided by the rrnDB database [22]. The average number of reads obtained after quality control was 126,378 (Appendix A).

***Genome-wide metabolic reconstruction***. To predict the metabolic potential of microbial taxa identified via 16S rRNA enumeration, we used a subsystem-based approach implemented in the microbial community of SEED, an application of the SEED genomic platform [23] as we have described previously [14]. Curated across human gut microbial genomes, metabolic subsystems include biochemical pathways classified into two categories: (i) the biosynthesis of vitamins, amino acids, and cofactors [24,25], and (ii) production of SCFAs [26]. To analyze the glycosyl hydrolase (GH) and polysaccharide lyase (PL) gene family abundance in profiled taxa, we obtained the distribution of carbohydrate active enzymes (CAZymes) in the analyzed reference genomes using the dbCAN2 tool [25] and observed the production of SCFAs. The obtained CAZyme family distribution was binarized, i.e., each genome and each family were assigned “1” if it contained at least one enzyme from this family; overwise, it was assigned “0”. The obtained binary phenotype matrix (BPM) for metabolic phenotypes and CAZyme family distributions in the reference genomes were used to calculate a community phenotype matrix for all mapped taxa obtained from 16S rRNA analysis as previously described [27]. To calculate the community phenotype index (CPI) for each metagenomic sample and each metabolic phenotype and GH/PL family, we used a development version of the Phenotype Profiler tool provided by PhenoBiome Inc. (Walnut Creek, CA, USA), as previously described [28].

***Statistical analyses.*** Differences in the Shannon alpha diversity measurements, community phenotypes including biosynthetic and degradation pathways, and the relative abundance of particular gut microbial taxa in experimental compared to control cultures were assessed using a double-tailed Mann–Whitney U test or t-test. Significant differences in the frequencies of taxa present between groups were assessed using a double-tailed chi-squared test.

***Data availability.*** 16S rRNA sequence data are available under BioProject accession: PRJNA927035. The NCBI provided a link for reviewers to access data. https://dataview.ncbi.nlm.nih.gov/object/PRJNA927035?reviewer=jo1i9a9jar5hdvg5dpgo6kcfvd; accessed 15 January 2023.

## 3. Results

### 3.1. Modulatory Potential of Prebiotics and Medicinal Herbs

We generated a fecal pool that combined an equal mass of stool from 12 healthy vegetarian human subjects used to inoculate a chemically defined medium (CDM) containing no carbohydrate substrates or a CDM supplemented with one of eleven prebiotic fibers or porcine gastric mucin. These were compared to a diverse group of 17 medicinal herbs used for digestive support, nootropics, immune modulation, and as culinary spices. Among the prebiotic fibers analyzed, most consist of glucose polymers that share common sugar linkages but may differ in the plant source, branch linkage, or density (Table 1). We included inulin, β-glucan, and FOS for comparison to assess the impact of altered sugar composition and sugar linkages and finally porcine gastric mucin that has a defined and more complex sugar composition, sugar linkages, undefined chain lengths, and branching density. By contrast, medicinal herbs are poorly defined with respect to these features and are mostly unknown. Previously, we measured the sugar composition of medicinal herbs via HPLC, confirming the presence of numerous sugar moieties, glucose as the dominant sugar, and varying amounts of glucuronic acid, galacturonic acid, ribose, xylose, galactose, arabinose and rhamnose, whereas mannose, glucosamine, and fucose were present in only some of the herbs tested [14,19], confirming the increased diversity of sugar moieties and assumed sugar linkages. Here, we investigated how known similarities and differences in glycan composition and structure influenced microbiota structure and community metabolism.

### 3.2. Modulatory Capacity of Prebiotic Fibers and Medicinal Herbs

We sequenced the V3–V4 region of the 16S rRNA gene derived from human fecal cultures. These sequences were enumerated at multiple levels to enable approximate species-level comparisons. Instances where precise species designation was not possible, multiple species of equal sequence identity are shown (Appendix A). In total, we observed 401 unique phylotypes and an average of 186 phylotypes in each unique culture condition. We calculated the Shannon alpha diversity of the control as well as the prebiotic and medicinal herb-supplemented communities (Figure 1A). Overall, the α diversity of cultures supplemented with prebiotics was lower than that of medicinal herbs, although citrus and apple pectin and corn dextrin generated high diversity values similar to those observed for medicinal herbs. Only α cellulose- and FOS-supplemented communities were of lower diversity compared to unsupplemented control cultures.

We performed Bray–Curtis and PCoA analysis based on the β diversity of the control and prebiotic-supplemented communities to determine how structural and compositional differences in prebiotic fibers altered the composition of fecal communities. This revealed that chemically and structurally related prebiotics selected communities with higher similarity (Figure 1B). Amylose and α cellulose are the simplest glycans analyzed and are comprised of linear glucose polymers with α 1-4 linkages. Cultures supplemented with these prebiotic fibers formed a loose cluster. Amylopectin is also comprised of linear glucose polymers with α 1-4 linkages with α 1-6 branching. Cultures supplemented with this prebiotic clustered closely with amylose and extended the first cluster. Corn and potato starch are glycans with varying proportions of amylose (~20%) and amylopectin (~80%). As expected, cultures supplemented with these prebiotics clustered together and further extended the first cluster, although technical replicates displayed some outliers. Corn dextrin is hydrolyzed with starch polymers of a lower molecular weight and is comprised of α 1-4 linkages with α 1-6 branching. Somewhat surprisingly, cultures supplemented with corn dextrin clustered distinctly from corn- and potato starch-supplemented cultures. Apple and citrus pectin are comprised of α 1-4 linkages of galacturonic acid that may be esterified with methoxy or hydroxyl groups and may be acetylated or contain chain-disrupting rhamnose moieties. Cultures were supplemented with apple and citrus pectin clustered together and separated from prebiotics comprised of glucose polymers. Citrus pectin clustered most closely to the control cultures, despite displaying substantially higher alpha diversity. Similarly, β-glucan, a polymer of glucose with β-1-3 and β-1-4 linkages clustered uniquely was distinctly separated from other α-linked glucose polymers. Inulin is a polymer of fructosyl residues with β-2-1 linkages and a terminal α-1-2 linked glucose molecule. FOS has the same composition but is distinct in its reduced polymerization length. Cultures supplemented with either inulin or FOS formed related but distinct clusters that were separated from other clusters. Porcine gastric mucin is a glycoprotein (~80 carbohydrate, ~20% protein) consisting of *O-*linked sugars that include *N-*acetylgalactosamine, *N-*acetylglucosamine, fucose, galactose, and sialic acid. As expected, mucin supplemented cultures clustered uniquely from all simple prebiotic fibers.

To further assess the modulatory potential of each prebiotic fiber and medicinal herb, we compared the relative abundance of taxa from control cultures to that of the prebiotic or medicinal herb cultures and scored each as increased, unchanged, or reduced based on a five-fold cut-off (Figure 1C; Appendix A). As expected, we observed that a larger number of taxa were altered in the medicinal herb-supplemented cultures compared to the defined prebiotics (ave = 190 vs. 100). Furthermore, the number of phylotypes observed in the herb-supplemented cultures was greater than those in the prebiotic-supplemented cultures (ave = 216 vs. 145). Apple pectin was the most modulatory prebiotic fiber (134 altered phylotypes), therefore being grouped at the low end of modulatory capacity, as with some medicinal herbs. Apple pectin supplemented cultures also displayed an increased number of phylotypes (ave = 196). Mucin, with an intermediate complexity of glycans, displayed a similar modulatory capacity (131 altered phylotypes) and supported a community of intermediate complexity (ave = 170 phylotypes). These results indicate that medicinal herbs support communities of increased complexity and display greater modulatory capacity compared to defined prebiotic fibers.

### 3.3. Taxonomic Analysis of Prebiotic and Herb-Driven Impact

We compared the relative abundance of families derived from cultures supplemented with prebiotic fiber and medicinal herbs compared to control cultures to identify patterns of responsiveness. Statistically significant changes are presented (Appendix A). We observed the comparable effects of prebiotic fibers and medicinal herbs in several instances. Among the most dominant families in control cultures were taxa belonging to Enterobacteriaceae. The relative abundance of this family was significantly decreased by six prebiotic fibers (50%) compared to the control cultures. Nine medicinal herbs (53%) resulted in significantly reduced relative abundance of Enterobacteriaceae. No treatments displayed significantly increased relative abundance. The dominant family, Lachnospiraceae were reduced in relative abundance by nine prebiotic fibers (75%) and nine medicinal herbs (53%). No treatments displayed significantly increased relative abundance. All prebiotic fibers except FOS (92%) significantly increased the relative abundance of Bacteroidaceae. Medicinal herbs also broadly increased this family (82%). No treatments displayed significantly decreased relative abundance.

We also observed differential responses between prebiotic and herb-treated cultures. The relative abundance of Acidaminococcaceae was increased by 13 medicinal herbs (~77%); only Triphala reduced the relative abundance. By contrast, nearly all prebiotic fibers (83%) reduced the relative abundance of this family. The relative abundance of Eggerthellaceae was selectively increased by potato (*p* = 0.004), corn starch (*p* = 0.002), corn amylopectin (*p* = 0.002), and amylose (*p* = 0.02) and decreased by corn dextrin (*p* = 0.002), FOS (*p* = 0.008) inulin (*p* = 0.004), and mucin (*p* = 0.002). Only two medicinal herbs (12%) increased the relative abundance of this family: licorice and slippery elm. Overwhelmingly, the relative abundance of this family was reduced by 14 medicinal herbs (82%). The family Bifidobacteriaceae has received much attention, and several reports indicate that selected prebiotics stimulate *Bifidobacterium* growth. Our data indicate that prebiotic fibers (42%) increased the relative abundance of this family, with these prebiotic fibers being corn amylopectin (*p* = 0.002), corn (*p* = 0.002), potato starch (*p* = 0.05), and apple and citrus pectin (*p* = 0.002). A larger proportion of medicinal herbs (71%) positively modulated this family. No treatments resulted in a significantly decreased relative abundance. Similar profiles were observed for the phylogenetically related Coriobacteriaceae. Peptostreptococcaceae were reduced in relative abundance in response to prebiotic fibers such as corn amylopectin (*p* = 0.02), citrus and apple pectin (*p* = 0.02; *p* = 0.002), β glucan (*p* = 0.002), FOS (*p* = 0.004), potato and corn starch (*p* = 0.004; *p* = 0.002), amylose (*p* = 0.002), and inulin (*p* = 0.002). This contrasts with medicinal herbs that displayed nominal effects; Triphala significantly increased relative abundance (*p* = 0.008) and slippery elm and licorice reduced the relative abundance of this family (*p* = 0.002 and *p* = 0.04, respectively).We examined the profiles at the genus and species level to identify patterns unique to prebiotic fibers or medicinal herbs as well as those taxa that are generally responsive to all treatments (Appendix A). A group of unresolved sequences related to *Absiella dolichum* and *Longicatena caecimuris* were preferentially increased by prebiotic fibers (4 instances, 33%), indicating the preference for α 1-4 and α 1-6 glucose polymers and *O*-linked sugars. Among the medicinal herbs, only Triphala-supplementation led to an increased relative abundance. A phylotype group with similarity to *A. dolichum* was also increased by three prebiotic fibers (25%): β glucan, corn starch, and inulin. A large spectrum of 13 herbs (77%), but not prebiotics, increased the relative abundance of *Acidaminococcus intestini.* This species was not detected in the control cultures but increased in relative abundance to varying extents in response to medicinal herbs. Similar profiles were observed for *Agathobaculum* spp. (formerly *Eubacterium*), and *Alistipes* spp. Were positively increased by medicinal herbs and mucin but not by prebiotic fibers. Interestingly, the relative abundance of *A. finegoldii, A. ihumii, A. indistinctus, A. obesi*, and *A. putredinis* were exclusively increased by medicinal herbs. *A. onderdonkii* and *A. shahii* displayed mixed increased/decreased responses to medicinal herbs, although in the case of *A. shahii,* only 2 (11%) were positively altered compared to 12 (71%) displaying reduced relative abundance. A poorly resolved cluster of species including *Aminipila butyrica*, *Emergencia timonensis*, *Ihubacter massiliensis*, and *Anaerovorax odorimutans* was exclusively modulated by medicinal herbs, 90% of which resulted in an increased relative abundance.

*Bacteroides* spp. were responsive to both medicinal herbs and prebiotic fibers. We profiled 25 distinct *Bacteroides* species groups with 29 unique treatments (725 unique combinations). We observed 328 statistically significant changes (45%). Medicinal herbs increased the relative abundance of *Bacteroides* spp. in 203 instances (48%) compared to 109 instances (36%) with prebiotic fibers and mucin. Prebiotics reduced the relative abundance of *Bacteroides* in only 10 instances (3%), compared to 6 instances (~1%) for medicinal herbs. Only *B. eggerthii* displayed a herb-specific response, increasing relative abundance in 35% of the herbs tested. Some taxa displaying a high frequency of increased representation in response to medicinal herbs were also highly responsive to prebiotics, including a species group with similarity to *B. faecis* and *B. thetaiotaomicron* (88% responded to herbs, 83% to prebiotics) and *B. xylanisolvens* (88% and 75%, respectively). Some *Bacteroides* spp. displayed preference for medicinal herbs compared to prebiotic fibers, including *B. doreii* (82% and 33%, respectively), two species clusters with similarity to *B. vulgatus* (100% and 42%, respectively), another unresolved group with similarity to *B. doreii* and *B. vulgatus* (100% and 58%, respectively), and *B. thetaiotaomicron* (82% and 50%, respectively). Conversely, some *Bacteroides* spp. preferentially expanded representation in response to prebiotic fibers such as *B. nordii* (6% and 67%, respectively) and *B. uniformis* (12% and 58%, respectively). These results are consistent with the high number and diversity of sugar catabolism functions encoded in *Bacteroides* genomes. The related genus *Parabacteroides* displayed greater overall preference for medicinal herbs compared to *Bacteroides* spp. All significant changes observed involved increased relative abundance compared to the control cultures. *P. distasonis* responded at a similar frequency to herbs and prebiotics (59% and 50%, respectively), whereas *P. goldsteinii* (94% and 25%, respectively), *P. johnsonii* (82% and 25%, respectively), and *P. merdae* (82% and 17%, respectively) displayed preference for medicinal herbs.

*Bifidobacterium,* another taxonomic group with extensive glycan degradation capacity, revealed five species groups responsive to treatments, but only *B. adolescentis* and *B. longum* were broadly responsive, each increasing the relative abundance to 59% of medicinal herbs and showing a 33% and 25% responsiveness to prebiotics, respectively. The results indicate a preference and/or capacity to utilize α 1-4 and α 1-6 glucose polymers and α 1-4 galacturonic acid polymers. The relative abundance of *Eggerthella* (due to *E. lenta* and *E. timonensis*) was selectively increased by amylopectin (*p* = 0.002), amylose (*p* = 0.02), and corn and potato starch (*p* = 0.002 and *p* = 0.004, respectively), indicating a preference for α 1-4, α 1-6 glucose polymers. No medicinal herbs increased the relative abundance of this genus. α-cellulose increased the relative abundance of *Escherichia* as the result of the increased representation of *E. coli* (*p* = 0.02)*,* whereas other prebiotics maintained similar or reduced levels, as observed in the control cultures. This was contrasted by the nearly universal decrease in the relative abundance in medicinal herb-supplemented cultures as well as the larger magnitude of change observed.

### 3.4. Prebiotic Specificity Determinants

To evaluate the extent that compositional and shared linkages of prebiotic fibers influenced the overlap or unique taxa positively modulated via treatment, we conducted a pairwise analysis of significantly altered taxa for each prebiotic pair. We calculated the number of occurrences wherein a pair of prebiotics induced the same bacterial species, the overlap (o), and the sum of the number of instances that taxa were significantly increased by each prebiotic alone (s). Since prebiotics or prebiotic pairs that enhanced the representation of large numbers of taxa would be expected to generate large (s) values, we established frequencies (f) for each prebiotic pair by calculating f% = o/s(×)100 for each prebiotic pair (Figure 2).

The prebiotic pair displaying the highest f-value, 36.5%, involved cultures supplemented with corn starch and amylopectin. This relationship is sensible given that corn starch is comprised of ~75% amylopectin and ~25% pectin, whereas the corn starch and amylose pair displayed a lower f-score, 22.5%. Similarly, the second highest pairwise relationship was observed in cultures supplemented with apple pectin and citrus pectin, f = 34.8%. FOS and inulin represented another positive control and generated an f-score of f = 30.7%. Conversely, we noted prebiotic pairs that are compositionally or structurally distinct with high f-values, including amylopectin paired with either apple or citrus pectin f = 28%. Among the 23 prebiotic pairs with low f-scores (<10), we observed an enrichment in pairs with different sugar compositions, including pairs involving mucin (6), FOS (6), and β-glucan (4). By contrast, some structurally related prebiotics resulted in low f-scores, including α cellulose paired with corn and potato starch (f = 5.5% and f = 8.6%, respectively), α cellulose paired with amylopectin and dextrin (f = 4.7% and f = 8.5%, respectively), and finally potato starch and dextrin (f = 7.4%).

### 3.5. Predicting Sugar Linkages in Medicinal Herbs

We calculated f-scores for each prebiotic–herb pair and observed a surprisingly coherent pattern (Appendix A). We focused on prebiotic–herb pairs with f-scores >10 and found that four prebiotics, apple pectin (ave = 21.1, range = 17.2–25.9), citrus pectin (ave = 18.2, range = 15.6–24), amylopectin (ave = 12.5, range = 10.1–22), and corn dextrin (ave = 15.5, range = 11.4–26.1) were observed when paired with all medicinal herbs (Appendix A). Several prebiotics never generated f-scores >10 when paired with any medicinal herb, including potato starch, FOS, and inulin. The remaining four prebiotics generated relatively low f-scores for a smaller number of medicinal herbs, and these were mucin, twelve herbs (ave = 12.3, range = 10.1–13.6), corn starch, three herbs (ave = 11.6, range = 10.1–14.4), β-glucan, three herbs (ave = 10.9, range = 10.5–11.3) and α cellulose, one herb (ave = 10.4). It is noteworthy that Triphala generated f-scores > 10 when paired with nine prebiotics, the largest number observed, consistent with Triphala, that is a formulation of three distinct herbs, and therefore likely to be the most complex among the medicinal herbs evaluated.

### 3.6. Glycosyl Hydrolase Restructuring by Prebiotic Glycans

The specificity of glycosyl hydrolases is poorly characterized and due to their massive diversification, they have remained elusive to large-scale substrate specificity studies. To gain potential insights into the selection and specificity of glycosyl hydrolases (GHs), we compared the relative abundance of 170 glycosyl hydrolases (GHs) and 33 polysaccharide lyases (PLs) families predicted to be encoded by communities following prebiotic stimulation. The community-wide sum of the positive and negative % increases/decreases for each prebiotic fiber differed widely (Figure 3A). The response driven by α cellulose and amylose (α 1-4 linkage) was positive but moderate in magnitude; however, related prebiotics containing α 1-4 linkage with α 1-6 branching displayed widely varying selection patterns, and both corn and potato starch negatively selected GH family sets, whereas dextrin strongly positively selected GH/PL family relative abundance. Interestingly, the prebiotic inducing the largest increases in the relative abundance of the GH and PL families was β-glucan. Apple pectin increased the greatest number of GH and PL families (162), whereas α cellulose had the smallest effect, increasing just 19 (Figure 3B). It is notable that these predicted effects on GH and PL family representation did not fully mirror the overall taxonomic modulatory potential of the tested prebiotic fibers (Figure 1C). Overall, the prebiotic fibers increased the relative abundance of 124 GH and 24 PL families (Appendix A). Eleven GH/PL families were increased by all prebiotics, whereas thirty-three were uniformly increased if α cellulose was removed from comparison. To determine whether the common composition and structure of the prebiotics selected specific sets of GH/PLs, we looked for patterns corresponding to the groupings that were shown (Table 1). Patterns of representation identified no cases wherein the substrate specificity could be discerned. For each of the GH/PLs analyzed, we observed on average that 7.2 of the 12 prebiotics significantly increased the relative abundance of the GH and PL families. This indicates that most positively selected GH/PL families responded to prebiotics spanning two or more groups (Table 1). Relatively few GHs (29) showed a uniform decreased representation, with an average of 6.4 prebiotics of 12 driving such effects. A total of 47 GHs and 11 PLs displayed mixed relative abundance (increased or decreased) in response to prebiotic fibers, with an average of 3.7 and 2.6 prebiotics/family, respectively. Among these functions, the overall response to prebiotics with identical sugar linkages did not display coherent alterations in GH representation.

### 3.7. Prebiotic Restructuring of Community Fermentation Pathways

The metabolism of glycans by gut microbes generates simpler sugars that can be transported into microbial cells that encode appropriate receptors and sugar metabolism pathways that ultimately generate ATP. The anaerobic fermentation of such sugars results in a number of products, including a series of SCFAs (acetate, propionate, butyrate, and lactate). The particular fermentation product is species-dependent. For example, *Bifidobacterium* predominantly generates lactate, whereas *Bacteroides* predominantly generate propionate. We used genome reconstruction using 2865 gut reference genomes [26] to identify taxa with the coding potential to generate major fermentation products. We previously reported herb-induced alterations of SCFAs [19], and so the focus here is on prebiotic fibers (Figure 4). Most of the prebiotics evaluated increased acetate, including amylose and potato starch (*p* = 0.02), apple pectin, citrus pectin, amylopectin, corn starch (*p* = 0.002), corn dextrin, FOS (*p* = 0.008), and inulin (*p* = 0.004). A similar number of prebiotics increased the representation of propionate producers, including inulin (*p* = 0.02), mucin (*p* = 0.04), β-glucan, corn dextrin, apple pectin (0.002), and citrus pectin (*p* = 0.009). Surprisingly, only mucin (*p* = 0.02)-supplemented cultures increased the relative abundance of butyrate producers. Only four prebiotics significantly impacted either D- or L-lactate producers. β-glucan increased D-lactate (*p* = 0.002) but decreased L-lactate (*p* = 0.002) producers. Amylopectin (*p* = 0.02), corn starch (*p* = 0.002), and potato starch (*p* = 0.03) all decreased D-lactate producer representation. Mucin-supplemented cultures (*p* = 0.03) increased L-lactate producer representation, whereas corn dextrin and FOS decreased L-lactate producers (*p* = 0.008). Interestingly, 10 prebiotics reduced the representation of ethanol producers, whereas none led to an increase. Finally, prebiotics had mixed results in the representation of formate producers. Amylose (*p* = 0.04), amylopectin (*p* = 0.004), corn starch (*p* = 0.002), and potato starch (*p* = 0.004) reduced the relative abundance of formate producers, whereas α cellulose (*p* = 0.004), β-glucan (*p* = 0.009), citrus pectin, corn dextrin, mucin (*p* = 0.002), FOS, and inulin (*p* = 0.004) increased the representation of formate producers.

### 3.8. Glycan-Induced Changes in Vitamin Biosynthetic Potential

We used genome reconstruction to evaluate the impact of prebiotics on the B-vitamin biosynthetic potential of the studied microbial communities. Among the prebiotic fibers that resulted in statistically significant changes in the vitamin B1 potential, apple (*p* = 0.002) and citrus pectin (*p* = 0.004), amylopectin (*p* = 0.04), corn starch (*p* = 0.04), and mucin (*p* = 0.04) all reduced the relative abundance of this pathway (Figure 5). The influence of prebiotics on vitamin B2 was mixed; dextrin (*p* = 0.008), FOS (*p* = 0.008) and inulin (*p* = 0.004) increased representation of this pathway, whereas amylopectin (*p* = 0.008), corn starch (*p* = 0.002), and potato starch (*p* = 0.004) reduced the relative abundance of genes encoding B2 biosynthesis. The relative abundance of the vitamin B3 pathways was increased by α cellulose (*p* = 0.009), β glucan (*p* = 0.002), and dextrin (*p* = 0.002), whereas apple pectin (*p* = 0.04), amylopectin (*p* = 0.004), corn (*p* = 0.002), and potato starch (*p* = 0.008) reduced the relative abundance of B3 biosynthetic potential. β glucan (*p* = 0.01), dextrin (*p* = 0.004), and FOS (*p* = 0.01) increased the representation of vitamin B5 pathways. By contrast, gastric mucin (*p* = 0.02) reduced this pathway. β glucan (*p* = 0.008), amylopectin (*p* = 0.01), and dextrin (*p* = 0.002) increased the representation on vitamin B6, whereas mucin (*p* = 0.004) reduced the relative abundance of this pathway. Only β glucan (*p* = 0.02) and FOS (*p* = 0.02) increased the relative abundance of vitamin B7, whereas apple pectin (*p* = 0.03), corn starch (*p* = 0.0.002) and gastric mucin (*p* = 0.002) reduced this pathway. Apple (*p* = 0.002) and citrus pectin (*p* = 0.002), inulin (*p* = 0.03) and gastric mucin (*p* = 0.002) all drove reduction in the relative abundance of vitamin B9 pathway potential, no prebiotics increased this pathway. Similarly, β glucan (*p* = 0.002), amylopectin (*p* = 0.002), FOS (*p* = 0.03), and inulin (*p* = 0.01) all reduced the relative abundance of the vitamin B12 coding potential. Vitamin K producers were increased by β glucan (*p* = 0.03), FOS (*p* = 0.02), inulin (*p* = 0.02), and potato starch (*p* = 0.03) and decreased by apple pectin (*p* = 0.04) and gastric mucin (*p* = 0.002). Queuosine (Q) synthesis pathways were enriched by β glucan (*p* = 0.002), dextrin (*p* = 0.02), and FOS (*p* = 0.008) and reduced by apple pectin (*p* = 0.04), amylopectin (*p* = 0.04), corn (*p* = 0.002), potato starch (*p* = 0.03), and gastric mucin (*p* = 0.02). Finally, β glucan (*p* = 0.004), dextrin (*p* = 0.03), and FOS (*p* = 0.008) increased lipoic acid producers, whereas corn starch (*p* = 0.002)-supplemented cultures displayed a reduction of this pathway.

## 4. Discussion

Numerous studies have analyzed the modulatory effects of prebiotic fibers, but far fewer have examined medicinal herbs. While defining healthy gut microbiota remains an elusive goal, research efforts have begun to establish beneficial microbes in particular contexts, such as their responsiveness to immunomodulatory cancer therapies [29,30] and the severity of COVID-19 infection [31]. In this regard, prebiotics and medicinal herbs represent a potentially powerful tool to modulate gut microbiota to achieve desired community structures. This study addressed two goals. The first was to examine the influence of sugar composition and sugar linkages on microbiota responsiveness. The second was to compare and contrast defined prebiotic fibers and undefined medicinal herbs with respect to their influence on the gut microbiota structure and functional potential. We selected prebiotics with identical and dissimilar sugar composition and similar sugar linkages to determine the extent that those properties dictate their influence on gut community structure.

### 4.1. Modulatory Capacity of Prebiotic Fibers and Medicinal Herbs

In the majority of cases, culturing fecal communities in media supplemented with medicinal herbs generated higher α diversity compared to prebiotic fibers (Figure 1A). This result was expected based on the assumption that a greater diversity of sugars and sugar linkages, among other characteristics inherent to medicinal herbs, would promote a higher diversity than with prebiotic fibers. The PCoA clustering of microbial communities revealed that prebiotic fibers of a similar sugar content and, more importantly, identical sugar linkages, displayed similar β-diversity (Figure 1B). Prebiotics featuring glucose residues with α 1,4 linkage clustered together and in turn clustered with communities selected by prebiotics that featured glucose residues with α 1,4 linkage with α 1,6 branching. Interestingly, dextrin derived from corn clustered separately from other prebiotics with α 1,4 linkage with α 1,6 branching. Dextrin is a hydrolysate of starch with reduced molecular weight that may positively or negatively alter its accessibility to catabolic processes. This explanation may also apply to the related but distinct clustering of inulin- and FOS-supplemented cultures. Three prebiotics (inulin, FOS, and β glucan) containing beta linkages clustered separately from those featuring α linkages. The results shown in Figure 1A were also consistent with measures of the modulatory potential based on the average change in the supplemented cultures relative to the controls (Figure 1C). Both the Shannon diversity index scores and modulatory potential cover a breadth from highest to lowest, potentially providing a rational basis for choosing modulatory substances to validate in vivo that are capable of driving the desired microbial community diversity and representation of key taxa of interest.

### 4.2. Taxonomic Analysis of Prebiotic and Herb-Driven Impact

Taxonomic analysis of 12 prebiotic fibers is difficult to summarize comprehensively; however, we focused on taxonomic features at the family, genus, and species levels that appeared to distinguish or be common to prebiotics and medicinal herbs. Common to about half of all prebiotics and medicinal herbs tested was the decreased relative abundance of Enterobacteriaceae, a family harboring pro-inflammatory species. Members of this family have been documented as elevated in a number of disease contexts, including obesity, IBD, and CRC [32,33]. Among the best studied beneficial taxa, members of Bifidobacteriaceae and the genus Bifidobacterium, we observed that medicinal herbs more frequently increased the relative abundance of this bacterial group compared to prebiotics. Surprisingly, we did not observe statistically significant increases in *Bifidobacterium* spp. in cultures supplemented with either inulin or FOS, as reported by numerous groups [34,35].

Taxonomic alterations unique to prebiotic fibers were less numerous than those that were herb-specific; however, certain examples were identified. The differential effects of prebiotic and herb-treated cultures on the relative abundance of Acidaminococcaceae is challenging to fully explain. Nearly all prebiotic fibers reduced the relative abundance of this family. This effect is most likely due to the metabolism of this group that is capable of utilizing amino acids as a sole energy source and the fact that they encode few if any sugar utilization functions. The reduced abundance of this group is most likely due to the expansion of sugar-utilizing microbes. The positive effect of medicinal herbs is difficult to explain as the growth medium contains non-limiting amino acid supplies, thereby suggesting the presence of unknown factor(s) produced by herb-responsive microbes that enhance their fitness.

We observed that the genus *Eggerthella* was positively modulated only by select prebiotics, but not by any medicinal herbs. The clinical implications of this result are vague, but an increasing number of studies of member species indicate their broad metabolic capacity to biotransform molecules, including medications such as levadopa [36] and the cardiac drug digoxin [37]. *E. lenta* has been identified as associated with various conditions such as major depressive disorder [38] and others; however, cause and effect relationships have not been established. By contrast, *E. lenta* has been shown to activate Th17 cells in colitis [39].

### 4.3. Prebiotic Specificity Determinants

The results of PCoA strongly support that sugar composition and sugar linkage are determinants of community β-diversity. We analyzed taxa that were responsive to prebiotics to document that the number of common and unique taxa increased with pairs of prebiotics and to further explore these relationships. We observed cases wherein the identity of the sugar composition and sugar linkages generated high f-scores featuring a high number of overlapping responsive taxa (corn starch and amylopectin, apple and citrus pectin, and FOS and inulin) We noted several instances wherein apple and citrus pectin paired with either dextrin or amylopectin unexpectedly generated high f-scores. This result may highlight that the α 1-4 linkage, more so than the sugar itself, drives this relationship. We also observed that despite the shared properties of α cellulose paired with corn and potato starch, amylopectin and dextrin all resulted in low f-scores, whereas pairs within this group of prebiotics not involving α cellulose resulted in high f-scores. α cellulose-supplemented cultures displayed low diversity and modulatory strength and suggest that the poor solubility or accessibility of α cellulose confounds these findings. We observed low f-scores for the potato starch and dextrin pair. These results may highlight the role of polymer chain length as a determinant of the microbiota response. These results suggest that sugar composition and sugar linkages are determinants of the taxa they impact, but that additional features such as solubility and accessibility play a significant role in their prebiotic effect also.

### 4.4. Predicting Sugar Linkages in Medicinal Herbs

Apple and citrus pectin and amylopectin and corn dextrin paired with all medicinal herbs generated high f-scores. These results suggest the generalized presence of α 1,4 glucose and α 1,4 galacturonic acid polymers in medicinal herbs. It is further suggested that a diverse group of species may catabolize these substrates rather interchangeably. Conversely, neither FOS or inulin when paired with any medicinal herb generated low f-scores, suggesting that β-2-1 fructosyl and/or an α-1-2 glucose residues are rare or inaccessible in medicinal herbs. Finally, gastric mucin generated high f-scores when paired with most medicinal herbs, suggesting the *O*-linked sugars are prevalent in many medicinal herbs.

### 4.5. Glycosyl Hydrolase Restructuring by Prebiotic Glycans

Prebiotics had substantial modulatory effects on GH/PLs’ representation. Our attempt to discern potential substrate specificities of GHs and PLs was unsuccessful. There are several possible explanations for this outcome. First, the average GH/PL was increased by 60% of the prebiotics tested, thereby hampering our ability to observe prebiotic group-specific effects. Second, and perhaps more important, the relative abundance change induced by prebiotics of any GH/PL is linked to other GH/PLs encoded in the genome that in many cases number in the hundreds. This creates a large source of noise that prevent such signals from being identifiable. Transcriptomic approaches are better suited to addressing this open question. Similar negative outcomes were observed in our analysis of the sugar utilization pathway potential, which likely suffered from similar confounding effects.

### 4.6. Prebiotic Restructuring of Community Fermentation and Vitamin Pathways

Prebiotic-supplemented fecal cultures led to complex alterations and the restructuring of metabolic potential with respect to the sugar fermentation product and SCFA as well as the vitamin biosynthetic pathway representation. Overall, we were unable to discern any patterns between the prebiotics with similar composition and sugar linkages and the representation of fermentation pathways. Vitamin biosynthetic pathways were substantially altered via prebiotic treatments. Unexpectedly, prebiotics either tended to positively select for most vitamin pathways (α cellulose, β-glucan, dextrin, FOS, and inulin), or negatively select for most pathways (apple pectin, corn starch, mucin, and potato starch), whereas amylopectin, amylose, and citrus pectin resulted in a mixture of positive and negative selections (Figure 5). We were unable to discern any patterns between prebiotics with similar composition and sugar linkages and the representation of vitamin pathways.

### 4.7. Limitations of the Study

The results presented here are derived from in vitro culture. It is our expectation and experience that the community response to prebiotics and medicinal herbs in vivo does not perfectly align with those observed in vitro. However, in vitro culturing reveals the microbiological capacity of microbes to utilize prebiotic and herb substrates within a community and therefore represents useful information to further our understanding of complex biological interactions. An additional limitation of our study is the use of a single timepoint for the community analysis. It has been established that various factors, e.g., the pH as well as others, influence the community metabolism, suggesting that end-point analysis as presented here may not accurately reflect communities at earlier time points. Finally, we applied statistics that do not correct for multiple comparisons. Since we were testing a specific hypothesis, that sugar composition and linkage type are the primary drivers of microbiota response, together with the small (n) for each condition, we applied more relaxed statistical methods. Finally, a number of taxa were associated with a very low relative abundance (low raw read counts). We used pseudo values for taxa displaying zero values to enable ratios to be calculated. As such, these values and their ratios should be interpreted with caution.

## 5. Conclusions

A central focus of this study was to determine whether prebiotics with the same sugar composition and sugar linkages would impact various microbiome measures in a coherent manner. PCoA of communities selected by prebiotics mostly displayed the clustering of samples treated with related prebiotics. An examination of the prebiotic pairs and the taxa they co-increased provided further evidence that chemically and structurally related prebiotics considered as pairs frequently selected the same taxa. Given these results, it was surprising that we were unable to identify patterns in the taxa, GH/PLs, sugar utilization, fermentation products, or vitamin biosynthesis that correlated with the sugar composition or sugar linkages when considering prebiotics as larger related groups. These findings represent in vitro results and therefore are not expected to be fully reproducible in vivo. However, in vitro results are indicative of the microbial community’s capability to utilize prebiotics and medicinal herbs, and therefore they may be effectively used to serve as a prebiotic “menu” to select microbiota modulators best suited to achieve a specific configuration of interest.

## Figures and Tables

**Figure 1 microorganisms-11-00868-f001:**
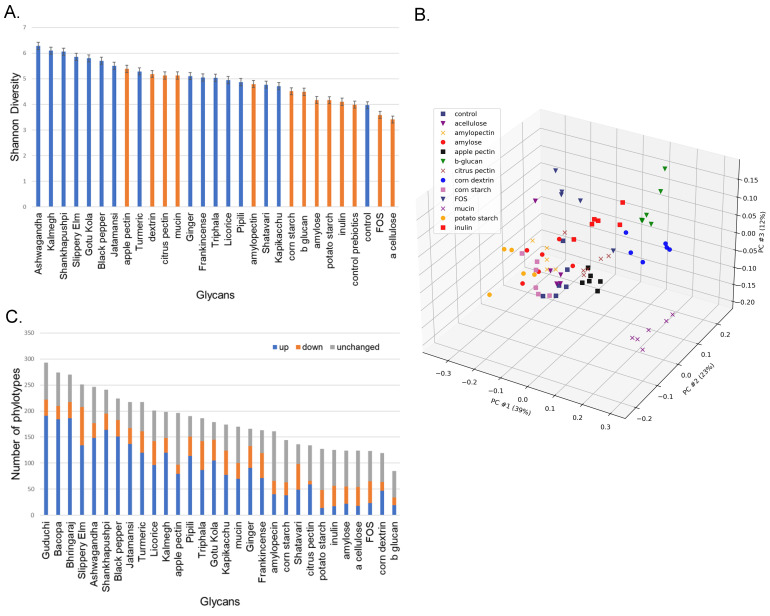
(**A**). Alpha diversity. Shannon diversity measures for control cultures compared to those observed in CDM supplemented with prebiotic fibers (orange) or medicinal herbs (blue). (**B**). Principal component analysis of prebiotic and medicinal herb-supplemented cultures. Bray–Curtis β-diversity measures of control cultures compared to those supplemented with prebiotics and medicinal herbs. (**C**). Comparison of modulatory effects. The average relative abundance of individual taxa in prebiotic and medicinal herb-supplemented cultures was compared to control cultures. Taxa displaying increased or decreased relative abundance (>5-fold) were summed and reported as altered.

**Figure 2 microorganisms-11-00868-f002:**
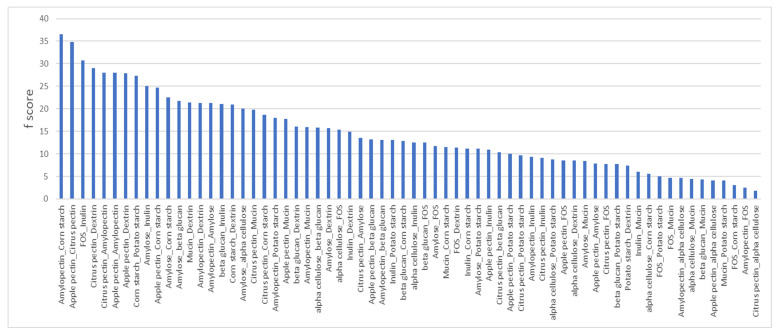
Pairwise analysis of prebiotics. The number of taxa impacted uniquely or that were in common for 66 prebiotic pair combinations. F-scores were calculated by dividing the sum of taxa impacted by prebiotics individually and dividing this figure by the number of taxa impacted by both.

**Figure 3 microorganisms-11-00868-f003:**
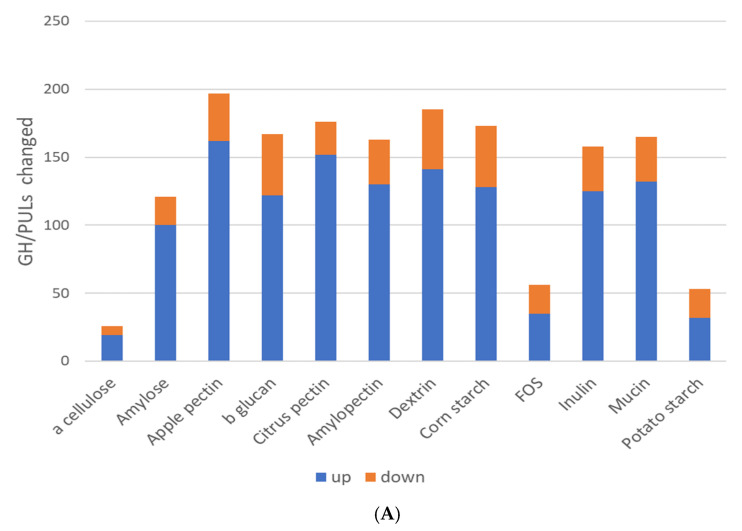
(**A**). Prebiotics alter CAZyme representation. GH and PL family representation were determined via genome reconstruction. The % increase or decrease in the prebiotic-supplemented cultures relative to the controls was determined. (**B**). The overall impact of prebiotics on GH/PLs representation. The % change in GH/PLs was determined by comparing the prebiotic to the control cultures. The % change for all GH/PLs was summed.

**Figure 4 microorganisms-11-00868-f004:**
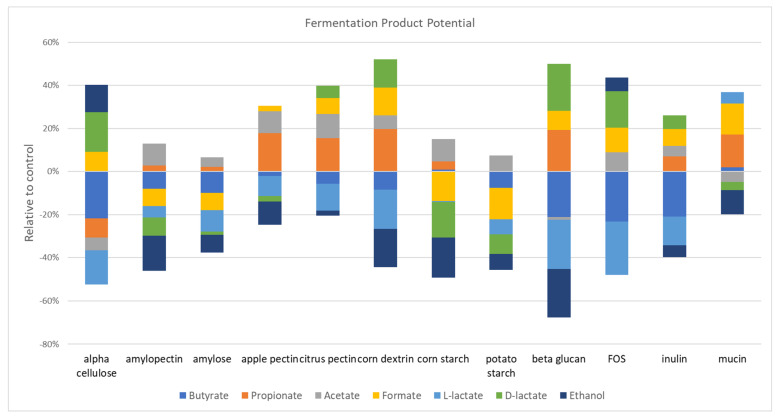
Prebiotic supplementation alters the representation of SCFA biosynthetic capacity. The impact of the relative abundance of communities stimulated with prebiotics on the predicted capacity to synthesize SCFAs and products of fermentation (colored bars).

**Figure 5 microorganisms-11-00868-f005:**
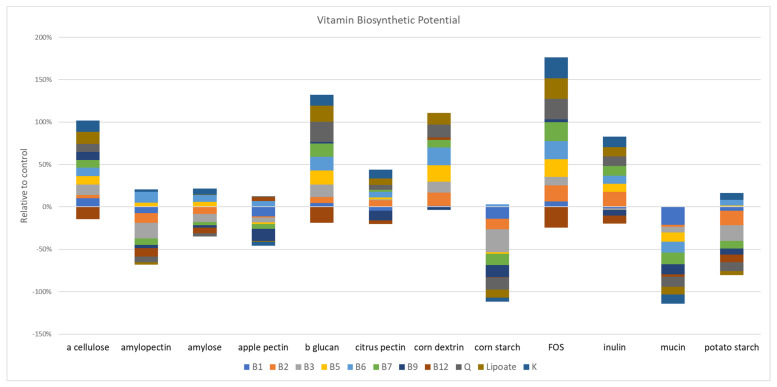
SCFA supplementation alters the representation of vitamin biosynthetic capacity. The impact oof the relative abundance of communities stimulated with prebiotics on the predicted capacity to synthesize vitamins (colored bars).

**Table 1 microorganisms-11-00868-t001:** Prebiotic composition and structure.

Prebiotic	Group	Sugar	Linkage
α cellulose	1	Glu	α-1-4
potato amylose	1	Glu	α-1-4
corn amylopectin	1a	Glu	α-1-4, α-1-6
corn dextrin	1a	Glu	α-1-4, α-1-6
corn starch	1a	Glu	α-1-4, α-1-6
potato starch	1a	Glu	α-1-4, α-1-6
apple pectin	2	GalA	α-1-4
citrus pectin	2	GalA	α-1-4
inulin	3	Fru	β-2-1
FOS	3	Fru	β-2-1
barley b-glucan	4	Glu	β-1-3, β-1-4
mucin	5	mixed	*O-*linked
Bhringaraj	NA	Rha, Ara, Gal, Glc, Man, Xyl, GalA, GlcA	ND
Guduchi	NA	Rha, Ara, Gal, Glc, Xyl, GalA, GlcA	ND
Bacopa	NA	Rha, Ara, Gal, Glc, Man, Xyl, GalA, GlcA	ND
Ashwagandha	NA	Rha, Ara, Gal, Glc, Man, Xyl, GalA, GlcA	ND
Kalmegh	NA	ND	ND
Shankhapushpi	NA	Rha, Ara, Gal, Glc, Man, Xyl, GalA, GlcA	ND
Slippery Elm	NA	ND	ND
Gotu Kola	NA	Rha, Ara, Gal, Glc, Man, Xyl, GalA, GlcA	ND
Black Pepper	NA	Fuc, Rha, Ara, Gal, Glc, Man, Xyl, Rib, GalA, GlcA	ND
Jatamansi	NA	Rha, Ara, Gal, Glc, Man, Xyl, GalA, GlcA	ND
Turmeric	NA	Fuc, Rha, Ara, Gal, Glc, Man, Xyl, Rib, GalA, GlcA	ND
Ginger	NA	Fuc, Rha, Ara, Gal, Glc, Man, Xyl, Rib, GalA, GlcA	ND
Frankincense	NA	Ara, Gal, Glc, Man, Xyl, GlcA	ND
Triphala	NA	ND	ND
Licorice	NA	ND	ND
Pipili	NA	Fuc, Rha, Ara, Gal, Glc, Man, Xyl, Rib, GalA, GlcA	ND
Shatavari	NA	Rha, Ara, Gal, Glc, Man, Xyl, GalA, GlcA	ND
Kapikacchu	NA	Rha, Ara, Gal, Glc, Man, Xyl, GlcA	ND

## Data Availability

16S rRNA sequence data are available under BioProject accession: PRJNA927035. The NCBI provided a link for reviewers to access data. https://www.ncbi.nlm.nih.gov/bioproject/PRJNA927035; accessed 13 February 2023. Additional data published previously may be found in BioProject IDs PRJNA545727, PRJNA497131, PRJNA632044.

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
