# Peer review of "Alteration of Community Metabolism by Prebiotics and Medicinal Herbs"

_microorganisms, 2023, doi:10.3390/microorganisms11040868_

Round 1

Reviewer 1 Report

Peterson et al. evaluated the effects of prebiotics and medicinal herbs in vitro. I only have a few comments on this manuscript.

1.     Line 141: ‘9% H2, 81% N2’? The composition of the left?

2.     Line 144: Please make a clear description of ‘HEPES’.

3.     Lines 157-159: Please add a table about the information of the medicinal herbs applied in the study.

4.     Line 165: typo.

5.     Lines 185-188: Please correct the font.

6.     Lines 213-214: As far as I known, it is hard to infer to species-level based on the V3-V4 region of the 16S rRNA gene.

7.     Line 218: I suggest to label the bars with different colors according to the supplements (prebiotics or medicinal herbs).

8.     Lines 218, 230, 265: Please merge the three panels into one single figure. In Figure 1C, number of taxa refers to OTU/ASV?

9.     Line 431: ‘a cellulose’? or ‘α cellulose’.

10.  Lines 461-472: It seems that the authors forget to add related results in the manuscript. I think the statements based on the ‘data not shown’ is not appropriate for the manuscript.

Reviewer 2 Report

Authors used a pooled stool sample from 12 vegetarian participants to evaluate how differences in glycan composition of prebiotic fibers and medicinal herbs alter the structure and metabolism of the microbiota. Overall, the paper is interesting but has many flaws, most of them significant. There are significant issues with the methods, sample size, and conclusions.

Comments

Title: Reprogramming of community metabolism by prebiotics and medicinal herbs. Reprogramming means a totally shift in the microbial community and this was not seen in the study. In addition, authors are comparing only one time point with the control, which is not a representative number.

Line 22. We applied genome reconstruction of enumerated communities to compare and contrast the structural and functional impact of prebiotics and medicinal herbs.

Genome reconstruction is when you do binning to recover MAGs from metagenomes. In this study, authors estimated or predicted composition and function from 16S rRNA sequences, which is no accurate because it is based in one single conserved sequence.

Methods:

How many reads were sequenced per sample?

Most of the abundance values listed in table 1 are very low. How do you verify that these values are real and no artifacts?

2.2. Anaerobic fecal cultures.

Line 140. Samples were grown statically for 2- 3 days at 37C as technical replicates (n=4-6) and grown to approximate saturation.

It isn’t clear when the samples were taken for sequencing. Please clarify.  

2.7. Statistical analyses.

When comparing the relative abundance of taxa between two groups, did you apply FDR correction? Or any multiple testing correction method?

Results

Figures 1A, 1B, and 1C should be merged into one single figure.

Figures 4 and 5difficult to read.

Line 165. 2.516. S rRNA sequence analysis. Rephrase

Discussion is long and redundant with the results.

Round 2

Reviewer 2 Report

Minor comments:

Methods

16S rRNA sequence analysis: be consistent between “16S rRNA sequence” and “16S sequence”

Line 200. Define HGM

Line 347. Be consistent between “herb supplemented cultures” and “herb-supplemented cultures”

Merge Figures 3A and 3B. Improve figure 3b, labels are overlapping with bars.

Line 529. Fix abundanceInterestingly

Supplemental material:

There are 5 tables s2 in the excel file and only one table s2 is described in the manuscript. There is one table s3 but there are two descriptions for this table in the manuscript. Please clarify.

Author Response

We have made all of the suggested minor edits throughout the manuscript and figures.  As suggested, we have merged Figure 3a-3b.